# Impact of the 2007–2008 United States Economic Crisis on Pet Ownership

**DOI:** 10.3390/ani12213010

**Published:** 2022-11-02

**Authors:** Pablo Crespo, Marco Faytong-Haro

**Affiliations:** 1Etsy, Inc., 117 Adams St, Brooklyn, NY 11201, USA; 2Sociology and Demography Department, The Pennsylvania State University, University Park, PA 16802, USA; 3Ecuadorian Development Research Lab., Daule 090656, Guayas, Ecuador; 4School of Health, Universidad de Especialidades Espíritu Santo, Samborondón 0901952, Guayas, Ecuador

**Keywords:** pet ownership; economic crisis, causal impact, time use; household size

## Abstract

**Simple Summary:**

Do recessions increase pet ownership? We look at pet ownership rates from 2003 to 2018. We compare the estimated real trend of pet ownership in American households to a scenario in which the 2007–2008 financial crisis did not occur. Our findings suggest that the financial crisis caused households to increase their pet ownership, especially for dogs and cats.

**Abstract:**

Limited literature explores the relationship between economic impacts and pet ownership. Do people have more pets as a result of economic crises? In the current study, we answer this question by looking at the time series of pet ownership and children present in U.S. households from 2003 to 2018. We utilize a causal inference technique to compare the estimated real trend of pet ownership in American households against the scenario in which the 2007–2008 financial crisis would not have occurred. Our findings suggest that the financial crisis triggered households to own more pets, specifically dogs and cats.

## 1. Introduction

Pets have become essential to many Americans’ lives [1]. Eighty-five million families, or 68% of U.S. households, own a pet [2]. According to the 2019/2020 National Pet Owners Survey, 84.9 million homes own a pet, equating to 67% of U.S. households. In 1988, this number was 56%. This signals a marked rise in the last 30 years. Dogs, cats, freshwater fish, and birds are the most common pets in U.S. households [2].

Limited literature has explored the idea that pet ownership has grown as child-rearing has declined in the United States [3], even during economic crises. A small study in an animal shelter in Chicago demonstrated that the 2007–2008 United States economic crisis influenced decreased pet adoption rates but did not affect pet relinquishment. The study also demonstrated that dogs were less adopted than cats, probably because dogs are more expensive to maintain [4]. Despite this, pet ownership rates increased after this event. There was a 6% point increase in pet ownership between 2008 and 2013 [5]. During the same years, the birth rate has slightly declined (in 2008, 13.8, and in 2013, 12.5) [6]. The trends are clear: people are having the same or fewer children and are owning more pets, which could mean they are opting more for pets than children.

One of the main reasons Americans may opt for pets instead of children is that the former are cheaper to support and do not require as much attention [7], which could matter more during an economic crisis. The yearly investment in a pet can range from $700 to $4000, including recurring medical costs [8,9]. Instead, raising a child until they finish secondary education is more costly, averaging $17,000 per year [10]. Emotional support given by animals could be a way to cope with financial distress, especially during crises [11]. Increased unemployment and leisure time are consequences of financial crisis events [12,13]. Therefore, there could be increased time to care for a companion animal. As adoption availability increases [4], people who want to adopt but cannot do it due to availability could be more likely to do so after the crisis [14].

### 1.1. Flexihood Personhood of Pets

Pets are a good source of companionship and tend to make people feel secure and relieve stress [15]. Indeed, around 90% of pet owners in the United States regard their pets as family members [16]. Recently, researchers talked about how people impute dogs and cats a “flexible personhood” [17]. Shirt-Vertesh [17] explains the following in their ethnographic study of pet owners in Israel: “pets are treated as flexible persons or emotional commodities; they are loved and incorporated into human lives but can at any moment be demoted and moved outside of the home and the family”. This premise could suggest that animals have the same adaptability as humans regarding acceptance and exclusion from the household. On the other hand, this adaptability appears to be more extreme, rapid, and problem-free for animals in comparison to humans. Furthermore, when dealing with animals, the degree to which kinship ties can be adjusted depends on whether animals receive this personhood. Pet-human kinship can be so strong that a recent study found that after a flood which is a situation of economic distress, the demand for pet-friendly housing lists was exceptionally high in comparison to other years [18].

### 1.2. Pets and Family Size

Empirically, minimal research has explored the relationship between pet ownership and family composition dynamics. For example, households in the United States with more children own and spend less on pets [19,20,21]. Decision-making power regarding family may also influence pet ownership. Hahn et al. [22] found that the probability of owning any pet was higher for women who lived in states where abortion was legal around 1970. These women had more control over their child-rearing expectations and, thus, their future family composition. Their study suggests that women have more pets when they have more decision power over their family composition. Instead, if they could not have abortions, they probably had to carry the baby to term and, therefore, were less likely to own a pet in the future. These two studies suggest a possible connection between family size and pet ownership.

### 1.3. Present Study

Given the growing pet markets in many developed countries, the relationship between economic impacts and pet ownership is essential to estimate the demand for pets and pet related products and services. To our knowledge, no studies explore variations in pet ownership during times of financial strain. In the present paper, employing collapsed data from the American Time Use Survey and causal inference techniques, we demonstrate that the 2007–2008 financial crisis in the United States had a positive causal effect on pet ownership and time spent with pets among childless households. Therefore, economic constraints galvanize pet ownership.

Finally, according to Zasloff [23], “a dog is not a cat, is not a bird”, indicating differing levels of pet popularity by species. Additionally, we use information from the American Pet Products Manufacturers Association [24] to estimate pet ownership trends by species. We expected that the species that increased the most were those with more “flexible personhood”, such as dogs and cats. Our findings reflect what we expected: dogs and cats have increased compared to the other species in the dataset (birds and fish) after the crisis.

## 2. Methodology

### 2.1. Datasets

We employ the American Time Use Survey (ATUS) data from 2003–2018 for the main study (the causal one) [25]. The sample consists of adults aged 18 years and up. The final analytic sample contains 201151 respondents (88,297 males and 112,854 females). The ATUS collected 1-day time diaries from respondents; as part of a telephone interview. Respondents were asked to recall the primary activity they were doing in each period, starting at 4 a.m. on the day before the interview and noting a start and end time for each activity up through 4 a.m. the day of the interview. While the most recent day may not be typical for any one respondent, especially for activities done only sporadically for the sample as a whole, this is a more accurate way to assess what average days look like than asking people what they typically do, which is likely to be more affected by both the lack of clear events memory and desirability bias. Although we only have time use data on one household member, this is sufficient to assess the average time spent with a pet for the average adult American.

The main reason why we chose ATUS for the principal causal analysis and not other datasets, such as the AVMA Animal Health Studies Database [26] or the Simmons Pet Food Statistics and Demographics Dataset [27], that have more specific questions about pet ownership information, was that ATUS is a genuinely representative sample of the American population and that does not bias the causal inference analysis. For the supplementary analysis of pet species trends, we employ data from the American Pet Products Manufacturers Association from 2000 to 2017.

### 2.2. Main Variable

First, we selected the ATUS activity “care for animals and pets (not veterinary care)” to construct our primary variable. Examples from the ATUS wording for this activity include caring for household pets, cleaning up after pets, watching kittens or puppies being born, feeding/watering pets, petting animals, clipping cat’s claws, adopting a pet, visiting an animal shelter to choose a pet, clip dog’s nails, bathing dogs, change dog’s water, feed rabbit, feed birds, groom pets, feed fish, grooming horses, feed guinea pig, and brushing dogs. These are unpaid activities; otherwise, they would be among the ATUS work activities.

Households in which their surveyed members spent time taking care of a pet were assigned as pet owners. Those who did not spend any time taking care of a pet were marked as non-owners. Therefore, we created a pet ownership proportion variable per year. The data for the primary causal analysis was constructed assuming that pet ownership is equivalent to having spent time taking care of pets in the household. This assumption has been previously used in other empirical studies present in the literature using the same survey [22,28,29].

### 2.3. Statistical Analysis

We present the yearly ownership proportion from 2003 to 2018. The economic crisis started in 2007, and the study period was divided into pre-crisis (2003–2007) and post-crisis periods (2008–2013). To investigate the causal impact of the financial crisis on pet ownership trends, we applied Bayesian structural time series models using the R-4.2.1 [30], CausalImpact [31]. The methodology employs one or several time series trends not affected by a specific treatment (an intervention) but with a trend moving similarly to the affected series before the treatment takes place. This condition must be met for the counterfactual series to be reasonable since fitting on divergent series would likely lead to bad fits in the pre-treatment stage and poor predictions in the post-treatment stage. These series are then fitted to the affected series as closely as possible in the pre-treatment period. The fitting model is then used with the values post-intervention to create a prediction of what the affected series would have looked like if the treatment had never taken place (i.e., a counterfactual). Any significant gap between the counterfactual and the observed series in the post-treatment period is thus attributable to the effect of the treatment itself.

In the case of the CausalImpact tool, the model fitting and creating the predictions iteratively in the post-intervention stage is a Bayesian regression with a spike-and-slab prior. This methodology iterates its estimates and updates, generating a distribution that allows the generation of confidence intervals around the counterfactual.

In our case, the treatment event is the financial crisis of 2007. The time series for the “treated” group is the time series of the proportion of households with pets where there are no children under the age of 18. The series used to create the counterfactual (“control”) is, in turn, the proportion of households with pets that also have children under the age of 18 in them. The latter was chosen because we do not expect the crisis to change the trend of pet ownership in this group. Furthermore, Figure 1 shows that the series moves in the same general direction as the treated series, making it a good candidate for generating a counterfactual.

The results are plotted in three panels. The first panel shows the data and a counterfactual prediction for the post-treatment period. The second panel shows the difference between observed data and counterfactual predictions; this difference is the pointwise causal effect, as estimated by the model. The third panel adds up the pointwise contributions from the second panel, resulting in a plot of the cumulative effect of the treatment.

## 3. Results

### 3.1. Trends in Average Minutes of Weekly Care for Pets and Household Members

To motivate the pet ownership analysis, Figure 2 shows the trends of average minutes of weekly care of pets and household members. The average baseline minutes of weekly care of pets was 5 in 2003 (pre-crisis period). After the economic crisis in 2007–2008, average pet care time increased to 7 min by 2018. On the other hand, the average weekly minutes spent on taking care of people inside the household was 33 in 2003 and increased to its peak of 36 in 2006. However, this average decreased from its peak beginning in 2008 to 34 and vastly decreased to 31 in 2010. Since then, it recovered slightly but has decreased since. It is noticeable that after the economic crisis, on average, the weekly minutes allocated to personal care decreased and have never returned to baseline averages before the crisis.

### 3.2. Causal Impact of 2007–2008 Economic Crisis on Pet Ownership

Now we turn to the causal impact Bayesian model in which our primary variable of interest is the yearly pet ownership proportion. This is shown in Figure 3. The Bayesian model showed that during the post-intervention period, pet ownership had an average value of 0.18. By contrast, without an intervention (crisis), we would have expected a pet ownership proportion of 0.16. The 95% interval of this counterfactual prediction is 0.16–0.17. Subtracting this prediction from the observed response yields an estimate of the intervention’s causal effect on the response variable. This effect is 0.021 with a 95% interval of 0.017–0.024. Summing up the individual data points during the post-intervention period, the pet ownership proportion had an overall value of 1.83. By contrast, had the intervention not taken place, we would have expected a sum of 1.63. The 95% interval of this prediction is 1.59–1.67.

The above results are given in terms of absolute numbers. In relative terms, the response variable showed an increase of +13%. The 95% interval of this percentage is [+10%, +15%]. This means that the positive effect observed during the intervention period is statistically significant and unlikely to be due to random fluctuations.

### 3.3. Robustness

As stated earlier, for robustness purposes, we used another indicator related to time spent with pets or pet owners to verify that the trend goes in a similar direction. Figure 4 shows from 1995 to 2015 the raw expenditures on pets per year for the United States (all the years were deflated). This trend is positive.

In other results not shown, we also separated the analysis by household size (2 and 3 or more). These robustness checks show that the positive effect of pet ownership post-crisis only holds for households of three members or fewer. This result suggests a semi-substitution between children and pets, as this probably affects families with only one child but, more specifically, with fewer family members.

### 3.4. Trends in Pet Species

Finally, as previously stated, our paper was interested in the causal increase of pets after the crisis and specific pet species acquisition trends. Figure 5 shows the number (in millions) of the four main pet species in the United States from 2000 to 2017. The graph shows that in 2000 there were 73 million cats in American households. In 2008 this number increased dramatically to 93.6 million, and in 2015 it slightly increased to 95.2 million. The number of dogs in American households was 68 million in 2000, 77.5 million in 2008, and 89.7 million in 2015, marking a stark increase after the crisis. While the trends are upward for dogs and cats, the opposite happens for birds and fish. There were 19 million pet birds in American households in 2000, 15 million in 2008, 14.3 million in 2014, and 20.6 million in 2015. While there is a positive difference of 1.6 million between the first and last years of information collected, the trend is anything but stable and upwards. In 2000 there were 159 million fish in American households, 171.7 million in 2008 (the only year this number increased in comparison to 2000), and 139.3 million in 2015. The graph shows a clear downward pattern of fish ownership in American households after the crisis.

## 4. Discussion

As we have demonstrated, pet ownership significantly increased in the United States after the crisis compared to a synthetic control counterfactual. To our knowledge, this is the first article that assesses how economic impact moves people to own pets and, more specifically, which specific species they own the most after the impact.

Using data from Statista and the APPA, we explored which pet species were most likely to be purchased after the economic crisis. The number of dogs increased the most in American households, followed by cats (net of possible decreased adoption rates). This finding holds if we link dogs and cats as a quasi-proxy for having children since they have been identified as having the most flexible personhood. However, after the crisis, there was a decrease in fish ownership, and bird ownership remained relatively stable. Compared to dogs and cats, fish and birds are assigned less flexible personhood; these results follow this line. Based on the limited data available, we can tentatively formulate the following relationship: when people experience financial stress, they are more likely to adopt pets.

Our results show that the 2007–2008 crisis catalyzed American households to increase their pet ownership. Also, the increase in pet expenditure at the national level in the same direction as the increase in pet ownership suggests that the relationship is robust.

One of the mechanisms behind the pet ownership increase could be that economic impacts make people rethink their child-rearing intentions and may own pets who act as “substitute children” (as raising pets is less costly than raising children), especially for adults without children of their own [32]. For example, Anderson [33] found that parrot owners often compare their pets to their children, leading some to coin the term “fids” to describe their “feathered kids”. Similarly, Greenebaum [34] conducted an ethnographic study of customers at a dog café in the United States and found that the majority of the owners treated their dogs like “fur babies”.

Additionally, pet-owning could be a mechanism to cope with the mental health consequences of economic crises. There is a wealth of evidence indicating that times of economic change are accompanied by increased risks of mental health diseases. People who are unemployed and living in poverty are at a much higher risk of developing mental health issues like depression, alcohol use disorders, and suicidal thoughts than those who have not experienced these hardships [35]. Pet owning could be a strategy to cope with these issues. Companion animals are increasingly gaining public recognition as a way to deal with stress and daily issues [36,37].

The pet-human well-being claim is supported by growing research. Pet ownership has been linked to decreased blood pressure, loneliness, anxiety, and fear [38,39]. Pet health-promoting benefits have been demonstrated in populations like the elderly [40] and children with serious illnesses [41,42]. Pets seem to play an especially significant role in the lives of people who are socially isolated or excluded, as they can provide solace, companionship, and a sense of worth. The bonds between people and animals can help smooth over life’s rough edges, providing fuel for perseverance [43], especially during financial crises. For example, a study in Malaysia demonstrated that during the COVID-lockdown, pet owners were found to have significantly higher levels of coping self-efficacy, positive emotions, and psychological well-being in comparison to non-pet owners [44].

This study does not come without limitations. The data for the primary causal analysis was constructed assuming that pet ownership is equivalent to having spent time taking care of pets in the household. While this could be a good approximation of pet ownership only and not more complex pet-human relationships, this highlights the need for a yearly nationwide representative study that includes pet ownership variables. The General Social Surveys (GSS) in the United States began to collect information on pet ownership in 2018 [45]. Since the GSS is a biannual effort, we think this is a promising method to explore further the relationship that pets have with other social forces at the national level and across time. Also, the APPA survey is not ideal; it is not nationally representative of the population employing a random sample [24].

Moreover, other social factors like declining child-rearing desires could be confounding factors for the increase in pet ownership. While this is possible, the increase in pet care time and decrease in household members’ care time after the crisis are stark and do not follow the same speed as their previous trends. Further studies could explore more deeply the mechanisms behind the increase in pet ownership during the 2007–2008 economic crisis, such as increased emotional support and unemployment, and leisure time.

## 5. Conclusions

Has pet ownership increased due to the 2007–2008 American economic crisis? In this study, we investigate this question using a causal inference method. We examine the estimated real trend of pet ownership in American households in the light of a hypothetical alternative in which the financial crisis of 2007–2008 never occurred. Our research suggests that the economic downturn encouraged families to increase their pet ownership, especially those of dogs and cats.

## Figures and Tables

**Figure 1 animals-12-03010-f001:**
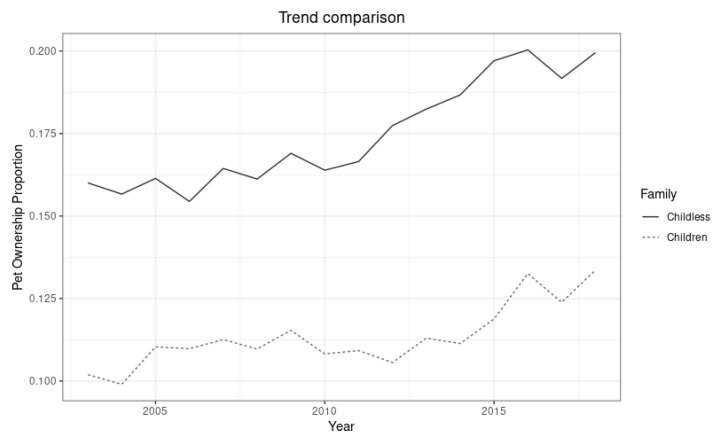
Trends of pet ownership in households with and without children.

**Figure 2 animals-12-03010-f002:**
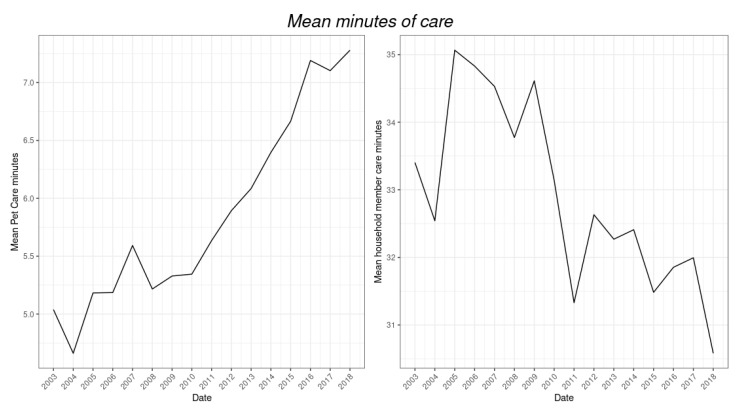
Average minutes of care weekly for pets (**left**) and household members (**right**).

**Figure 3 animals-12-03010-f003:**
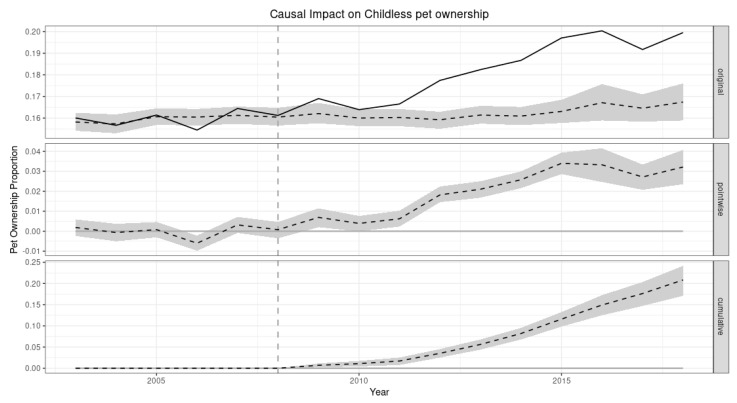
Causal impact on childless pet ownership. The first panel shows the actual data (solid line) and a prediction for what would have happened if the crisis had not occurred (dotted line). The second panel shows the pointwise causal effect, which is the difference between the actual data and the counterfactual predictions estimated by the model (dotted line). The third panel shows the cumulative pointwise causal effect (dotted line). The grey area refers to the 95% standard deviation boundaries.

**Figure 4 animals-12-03010-f004:**
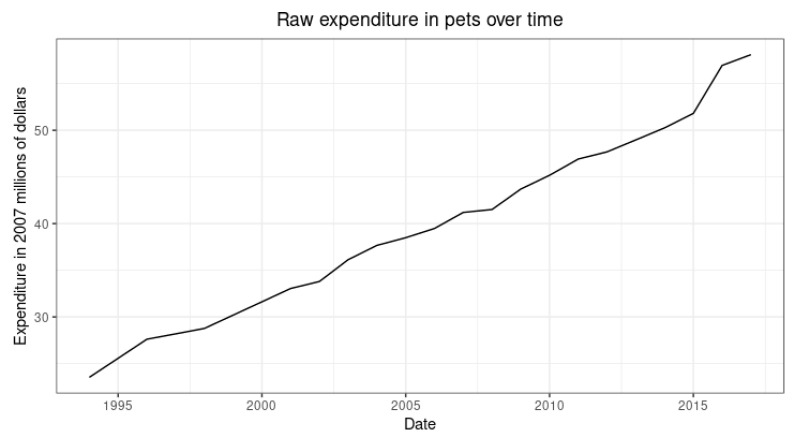
Pet Expenditures over time.

**Figure 5 animals-12-03010-f005:**
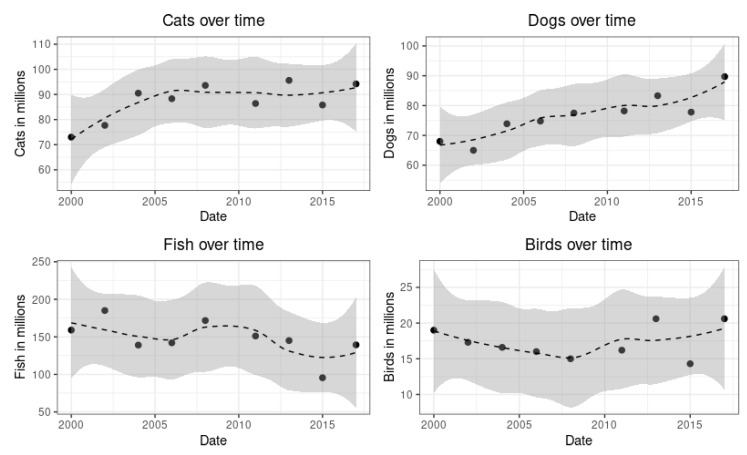
Pet counts over time.

## Data Availability

Publicly available datasets were analyzed in this study. This data can be found here: https://www.bls.gov/tus/data.htm (accessed on 10 October 2022).

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
