# Peer review of "Impact of the 2007–2008 United States Economic Crisis on Pet Ownership"

_animals, 2022, doi:10.3390/ani12213010_

Round 1

Reviewer 1 Report

The main question addressed by the research is whether or not people acquire more pets when the economy is bad

The topic is original.

The methodology seems correct, but it is not my field

 The authors assumed that people would have babies if they had not gotten a pet. To prove this point, they should get the statistics on the cost per year of a child and a cat and of a dog. A dog with proper veterinary care may be as costly as a child, at least until the child goes to college.

It might be interesting to add the shelter intake figures for those years. I believe more dogs and cats were relinquished during the recession.

It is interesting that people did not acquire horses during the recession and, in fact, the horse population of the US has not recovered.

The conclusions seem valid.

50 This statement seems contradictory. If a woman cannot have an abortion, she has more pets?

106 omit paid

Figure three Insert a legend, what is the solid line? What is the dashed line and shaded are area

Author Response

Responses to Reviewers

We are very grateful to the editor and the reviewers for providing comments on the manuscript. We have replied or/and addressed all the comments. Below are the specific comments provided (in bold) and our response (in regular font). The page numbers refer to the ones in the revised manuscript.

Thanks for your comments. We greatly appreciate them. We believe that the manuscript has improved significantly because of all the changes in the revision.

Point-by-point response

Reviewer 1:

The main question addressed by the research is whether or not people acquire more pets when the economy is bad. The topic is original. The methodology seems correct, but it is not my field (…) The conclusions seem valid.

1.     The authors assumed that people would have babies if they had not gotten a pet. To prove this point, they should get the statistics on the cost per year of a child and a cat and of a dog. A dog with proper veterinary care may be as costly as a child, at least until the child goes to college.

Thank you to Reviewer #1 for this comment. We followed this suggestion and included it in our introduction.

“One of the main reasons Americans may opt for pets instead of children is that the former are cheaper to support and do not require as much attention [6], which could matter more during an economic crisis. The yearly investment in a pet can range from $700 to $4000 including recurring medical costs [7,8]. Instead, raising a child until they finish secondary education is more costly, averaging $17000 per year [9]. Emotional support  given by animals could be a way to cope with financial distress, specially during crises [10]. Increased unemployment and increased leisure time are consequences of financial crisis events [11,12]. Therefore, there could be increased time to provide care to a companion animal. As adoption availability increases [4], people who wanted to adopt but could not do it due to availability could be more likely to do so after the crisis [13].”

2.     It might be interesting to add the shelter intake figures for those years. I believe more dogs and cats were relinquished during the recession.

Thank you to Reviewer #1 for this comment. We did not find reliable and clear adoption and relinquishment figures for our study period. While we found some figures and numbers, we did not think they were either nationally representative or collected appropriately. We did find a small study about a shelter in Chicago. We incorporated this to the introduction

A small study in an animal shelter in Chicago demonstrated that the 2007-2008 United States economic crisis influenced decreased pet adoption rated but did not affect pet relinquishment. The study also demonstrated that dogs were less adopted than cats, probably because dogs are more expensive to maintain [4].

3.     It is interesting that people did not acquire horses during the recession and, in fact, the horse population of the US has not recovered.

Thanks for this comment. What is happening with the horse population is really interesting. As horses are considered a special category of pets and our dataset does not specify horses, we could not include their trends in our analyses. However, we look forward to explore in the future the reasons of the horse population decrease.

4.     50 This statement seems contradictory. If a woman cannot have an abortion, she has more pets?

Thank you for this suggestion. We rewrote the paragraph so now the idea of that study is clearer.

5.     106 omit paid

Thank you. We followed this suggestion.

6.     Figure three Insert a legend, what is the solid line? What is the dashed line and shaded are

Thank you. Now the figure note has more details for the reader.

7.     Variable rating

Reviewer #1 rated in these item “Are the conclusions supported by results” .

as “Can be improved”. In this paper we opted for the style in which we detail much more the discussion and the conclusion is more succint (so the reader can grasp the one main idea of the paper in the conclusion). We followed the editor´s suggestions and included more detail to the introduction and to the discussion. We believe that the paper overall is now in better shape and it opens more doors to future research directions.

Reviewer 2 Report

Line 38/39 Two sentences do not make a paragraph

Line 45 - interpretation of quote not correct

Line 53 = missing a word in sentence

Line 66 -  This doesn't make sense.  How does "fertility flows" connect 

Line 52 - suggests a possible connection between 

Line 76 - pets and cats?

77 = compared to the species?

108 \ is it possible that women care for the pet and men would not respond that they spent time caring for the pet as the wife or female partner does so?

I think the literature review should include more about research done focusing on the idea that pets are substitutes for having children.  I know there has been recent studies in this area.  I'm not convinced that "fertility" is a good connection here.

Discussion should explore other variables that could be at play as reasons for why one would see an increase in time spent on pets after a financial crisis.  

Author Response

Responses to Reviewers

We are very grateful to the editor and the reviewers for providing comments on the manuscript. We have replied or/and addressed all the comments. Below are the specific comments provided (in bold) and our response (in regular font). The page numbers refer to the ones in the revised manuscript.

Thanks for your comments. We greatly appreciate them. We believe that the manuscript has improved significantly because of all the changes in the revision.

Point-by-point response

Reviewer #2:

1.     Line 38/39 Two sentences do not make a paragraph.

Thank you for this comment. We have corrected this.

2.     Line 45 - interpretation of quote not correct.

Thank you for this comment. We agree with this comment and have developed this paragraph with a more sound interpretation of Shirt­Vertesh´s quote.

3.     Line 53 = missing a word in sentence.

Thank you for this comment. We have corrected this.

4.     Line 66 -  This doesn't make sense.  How does "fertility flows" connect. 

Thank you for this comment. We apologize about that, it is probably the reminder ofan idea that was removed from that part before submission. We removed that sentence.

5.     Line 52 - suggests a possible connection between. 

Thank you to reviewer #2 for their suggestion. We modified the whole paragraph but also included the suggestion.

6.     Line 76 – pets and cats?

Thank you to reviewer #2 for their suggestion. We meant to say “dogs and cats”. That is corrected now.

7.     77 = compared to the species?

Thank you to reviewer #2 for their suggestion. We corrected the sentence: “Our results reflect what we expected: dogs and cats have increased compared to the other species in the dataset (birds and fish) after the crisis.”

8.     108 \ is it possible that women care for the pet and men would not respond that they spent time caring for the pet as the wife or female partner does so?

Thank you to reviewer #2 for their suggestion. To try to delve deeper into this comment, we first looked at the pet ownership rate by men and women in our study sample. Women own slightly more pets than men (16% versus 12%). We ran the causal analysis separately by sex and no differences were found in the causal effect so we decided to go on with our original analysis.

9.     I think the literature review should include more about research done focusing on the idea that pets are substitutes for having children.  I know there has been recent studies in this area.  I'm not convinced that "fertility" is a good connection here.

The editor suggested that, as well: to look for other explanations. Now the introduction is more consistent and details many other explanations. We do not detail them furthermore because the mechanisms behind why pet ownership increases more during the crisis is out of the scope of this paper and we cannot test them. We reflect more on the idea of why pets could be more needed during economic crises (costs, time, ecomotional support, etc).

10.  Discussion should explore other variables that could be at play as reasons for why one would see an increase in time spent on pets after a financial crisis.

The same as our answer in point 9, we discuss other reasons why this would happen.

11.  Variable rating

Reviewer #2 suggested that the introduction, cited references,  and support of conclusions could be improved. In this paper we opted for the style in which we detail much more the discussion and the conclusion is more succint (so the reader can grasp the one main idea of the paper in the conclusion). We followed the editor´s suggestions and included more detail to the introduction and to the discussion. We believe that the paper overall is now in better shape and it opens more doors to future research directions. Also, now there are more references that support our arguments.
